# Factors That Determine Innovation in Agrifood Firms

Juan Sebastián Castillo-Valero and María Carmen García-Cortijo *

Instituto de Desarrollo Regional, Universidad de Castilla-La Mancha, Campus Universitario s/n,
02071 Albacete, Spain; sebastian.castillo@uclm.es
* Correspondence: mariacarmen.gcortijo@uclm.es

**Abstract:** In this study, we aim to find the determinants of innovation in the agrifood industry in an inland region in southeast Spain, which depends upon and specializes in this sector. The determinants we propose are firm and environmental factors. From the empirical analyses based on Box–Cox models, we deduce that a firm's internal factors or characteristics are those that have the greatest influence on its propensity to innovate. Among them, firm size has the greatest effect. Innovation culture has the potential for exerting a multiplying effect via mechanisms such as knowledge spillovers or learning by doing.

**Keywords:** innovation; agrifood economy; innovation culture



## 1. Introduction

Innovation is one of the main determinants of productivity in agriculture and the agri-food sector [1–4], and is essential for increasing firms' competitiveness and economic growth [5–7].

However, the agri-food industry has certain special characteristics and, in spite of its importance in developed economies, has often been considered a low-technology industry [8]. It is therefore not possible to explain innovation using the traditional model based on science and technology [9–13]. An interactive approach is needed [14,15] to go beyond the classic or linear vision and delve into the organizational and systemic process involved in innovation by firms [15,16]. This approach must be systemic to find the factors that facilitate innovation behaviour [17].

Some authors focus on firms' internal factors [18–22], some on external or environ-mental factors [23–27], and others link both [28–31]. Some specific studies in the agrifood sector provide examples: (a) the study based on Dutch farmers which concludes that the most innovative are those that have the largest farms and reach a more heterogeneous market, in addition to having solvency and experience [20]; (b) the study that focuses on the European NUTS II-III regions which shows that innovative firms resort to external sources for research and are influenced by their socio-economic environment [26]; (c) a study that reveals that in Ireland there is a clear spatial concentration of agricultural innovation in areas with more research, education and advisory services [27].

Once the linear view of innovation had been set aside, the process was analysed as a complex phenomenon that depends on other agents, factors and the social structure of the environment [32–35]. The purpose of this document is to find out which factors help to explain innovation in the low-technology agri-food sector. We consider its main areas of activity: primary agricultural products, food processing and beverage processing. We focus specifically on a rural region in the southeast of Spain, Castilla-La Mancha. This is the second most important agricultural region in Spain, with agricultural production valued at 5.3753 billion euros [36]. Its agri-food industry accounts for 9% of the regional GDP, contributing 32% to total sales and employing 27% of the workforce [37]. Castilla-La Mancha and its agrifood sector are very much focused on the export market. In 2020, the sector increased its exports by 3% over 2019, reaching 2.644 billion euros, which amounted

to 37% of all the export revenue of Castilla-La Mancha. This figure is above the Spanish average of 21%. The goods that were exported most were beverages (wine), followed by meat products, fruit, vegetables and legumes, dairy products and eggs, and fats and oils. Another characteristic of Castilla-La Mancha exports is their dependence on European Union markets because the leading destinations are France, Germany, Italy, Portugal and the UK (which as from 2021 and Brexit will figure as non-EU). The first non-EU country, the USA, appears in the sixth position.

We therefore propose the following Hypothesis (H). Innovation by firms in the agrifood sector in Castilla-La Mancha depends on:

**Hypothesis 1 (H1).** *Firms' internal factors (economic performance, research personnel, legal form, age, size.*

**Hypothesis 2 (H2).** *External factors (rate of economic activity, population, research centres, level of education, location).*

*1.1. Internal Factors*

Firms' internal factors include business performance, because innovation activities entail high operation and maintenance costs which can only be covered by profitable firms as their greater resources allow them to take up potential market opportunities [38–40]. Some studies, however, claim that strengths in innovation in small enterprises are not based so much on the availability of resources as on knowledge that helps improve innovation capacity. This implies that initiatives to develop technological skills among entrepreneurs may generate positive external effects by also boosting the generation of new products [23].

Size is another factor that may affect the innovation process [41] although the literature on this is not conclusive [42]. For many authors [43–47], firm size is positively related to innovation while, for others [48–50], small firms are more flexible and adaptable and involve less bureaucracy so are better placed for R&D activities, which bring in revenue and allow them to occupy market niches that are not so attractive for large firms. Other authors find that the relation between firm size and innovation efficiency cannot be confirmed but, in general, small firms are in a worse position while medium-sized firms can obtain better results [51]. There are differential characteristics between innovative firms of different sizes. Some estimates stress that small firms stand out but in ways that vary depending on their sector [52].

Firm age should also be considered, because it determines the firm's experience and the resources available for adopting innovations [53–56]. However, some authors indicate that the link between firm age and innovation is negative, because firms become increasingly less able to generate new or important innovations as they grow older. Firms with greater experience find it difficult to stand up to constant external developments with the result that their products may become obsolete with respect to the latest environmental demands [57,58].

Another factor is the quality of the firm's human capital (human resources). A firm's innovation is based on its skill at using and developing the knowledge available in its environment [59] so the training profile of its labour force can contribute to its innovation capacity [60–63] or limit it [64]. Some authors, however, do not consider the link between the two variables to be conclusive [65]. Other studies find that training is needed in the agrifood sector for innovation to be applicable and deployable in the territory. Training is a tool that helps in generational change because it allows assets to be captured in increasingly diverse areas and sectors. Moreover, if the agricultural sector is to gain prestige and attract talent, training strategies with a medium and long-term perspective will be needed to develop such processes: so processes for generating and activating knowledge are becoming increasingly ubiquitous, varied and flexible [66].

Another aspect of interest that has not generally been covered by the literature is the firm's legal form, because usually no distinction is made between different types of firms and cooperatives. The presence of cooperatives in the production fabric is important for



the development of the innovation system from an evolutionary and territorial perspective [67]. It has been shown in developing countries that participation in cooperatives has a positive effect on the adoption of technology [68]. Cooperatives provide a mechanism for collective action that affects social responsibility by means of the share capital generated in relations between the cooperative and its stakeholders [69]. Share capital becomes an essential element and a strategic intangible asset that facilitates the process of innovation and provides the capabilities needed for the creation of knowledge and information—technical, organisational, commercial, financial, etc.; incorporating it in the firm's economic processes—design, production, distribution, etc.; and managing such processes to obtain greater innovation [70]. Since cooperatives are democratic organizations, that is, they are managed for and by their members, they promote the development of trust-based networks that foster learning at both organizational and collective levels, helping information to circulate and promoting incremental innovations [40,71–74]. Collective, participatory action feeds social innovation processes [75] and allows the partners to benefit from a multiplying effect as co-innovators [76].

### 1.2. External Factors

Apart from a firm's internal factors, innovation capacity can also be influenced by the firm's environment. Distinctions should be made in this dimension between economic activity, population, research centres, level of education, and location.

Economic activity and population can be considered as the assets of a region that distinguish it and allow it to be comparatively more competitive than other regions. Greater economic activity and greater population raise investment in science and technology and place firms in that area among the main generators of technology and new products and service by means of innovation [77,78].

The literature also points to the importance of external sources of knowledge from which a firm can benefit because of its location and proximity to innovation facilitators [30] by means of cooperation with other agents, such as universities, research centres, engineering firms, suppliers, customers or even competitors [79–81]. There have been studies that reveal the clear spatial concentration of innovation in agriculture in zones that have a higher distribution of research, education and advisory services, as discussed in the case of Ireland [27].

A firm's innovation capacity can be improved by the existence in its environment of research centres with which it can maintain permanent contact, forming values and social capital to increase trust and interaction among the various elements of the network [80,82]. Further, research centres and universities can offer a trained workforce as well as access to innovative processes and/or products [83]. Some studies propose an endogenous growth model in which innovation requires both researchers to generate inventions and entrepreneurs to adopt them. The problem arises when research and entrepreneurship compete for funds, in which case innovation has to be promoted through entrepreneurship rather than research [84].

Finally, location affects the scope and quality of innovation activities, which vary from region to region. We therefore consider it necessary to add the factor of territoriality when analysing the effect of the environment on an organization's propensity to innovate [31]. Urban areas with greater population density can support firms' innovation activity because of the proximity of research centres and universities [83]. Even so, the potential for rural innovation is very high and is promoted by a sound base of natural resources and community spirit, intelligent use of tacit knowledge and the use of cooperation and social innovation.

## 2. Materials and Methods

### 2.1. Measurement of Innovation

Although the goal of this document is to find the factors that best explain innovation, we first have to identify an indicator for it. Innovation in the agricultural sector is a complex process that is difficult to measure [3,35]. Some authors try to measure it using primary

data [85,86]; others use individual adoption of technology as a proxy for agricultural innovation [4,87], or secondary data to propose synthetic data for firms [88–92], while others build an index from indicators of the adoption of innovations, knowledge acquisition and permanent innovation [35].

Innovation is considered to be a process of significant change in a firm's product, process, marketing or organization [93]. Based on this premise, a variable for innovation is proposed based on the regional aid for innovation that firms apply for [92]. Such aid aims to promote R&D+I activities [93]. The same authors [92] add to the measurement the number of patents and utility models registered with the Spanish Patent and Trademark Office.

Although these authors treat the innovation index as a dichotomous variable, in order to give an analytical form to this concept of innovation, in this document we propose an index (II) for a firm (i), which takes its inspiration from the algorithm given in [94] but includes time-frequency (f) instead of frequency by activity and type of innovation. When the concept given by [92] is combined with the algorithm of [94], we get the following:

$$II_i = (f \text{ Focal aid } i)/N_1 + (f \text{ Innova aid } i)/N_2 + (f \text{ Patents } i)/N_3$$

where f is the frequency with which a firm i has applied for aid and/or has registered a patent. The time interval for the study is from 2008 to 2017, that is, 10 years. However, aid for innovation in Castilla-La Mancha was provided in different yearly periods, so $N_1$, $N_2$, $N_3$ are different: $N_1 = 9$, $N_2 = 7$ and $N_3 = 10$ (the only one covering the whole period). If we replace $N_1$, $N_2$, $N_3$ with their values, we get the following expression:

$$II_i = (f \text{ Focal aid } i)/9 + (f \text{ Innova aid } i)/7 + (f \text{ Patents } i)/10$$

Focal refers to aid covered by Rural Development Plans which are co-funded by the European Agricultural Fund for Rural Development (EAFRD) and national and regional Administrations. The aim is to increase the added value of agrifood firms by means of innovations in the processing and/or sale of their products. Innova refers to direct subsidies, 80% being funded by the EAFRD, which aim to promote R&D+I activities for the creation of new goods or services, or a significant improvement in existing ones and/or new or significantly enhanced processes. Finally, Patents and Utility Models refer to legally registered technological innovations (in our case, with the Spanish Patent and Trade Mark Office, SPTO).

The general average $II_i$ or the agri-food industry was 0.30, which indicates a low intensity of innovation [8,95–99]. By sector, the highest rate was for beverage processing firms, with II = 0.34; next was food processing with 0.31, then primary agricultural products with 0.24. These results are in line with national data, which show that investment in innovation in the primary agricultural products sector was 0.41% of sales volume, compared with 0.57% in the agri-food processing sector [100,101].

*2.2. Sample and Variables*

The database used in this study is made up of primary data from innovative agri-food firms in Castilla-La Mancha whose economic activity is classified as primary agricultural products, food processing and beverage processing. The study period is 2008 to 2017, which corresponds to two periods within the Rural Development Programme for Castilla-La Mancha. The total sample comprises 771 firms with II > 0 which represents all firms in Castilla-La Mancha which requested FOCAL and INNOVA aid during the period considered. We thus drew up a cross-section, and the variables used in this study are given in Tables 1 and 2.

**Table 1.** Definition of the dependent and independent variables.

| Variables | Typology | Description |
|---|---|---|
| | | **Dependent variable** |
| Innovation Index (II) | continuous | Source for FOCAL and INNOVA: Official Gazette of Castilla-La Mancha. Source for patents and utility models: Spanish Patent and Trademark Office |
| | | **Independent variables** |
| Performance (ROA) | discrete | Mean profitability from 2008 to 2017. It takes the value of 1 for firms located in the first quartile, 2 for those in the second quartile, and 3 for those in the third quartile. Source: Own calculation based on SABI |
| Size (SIZE) | discrete | Size of the company. It takes the value of 1 if it is a micro-enterprise, 2 if it is small, 3 if it is medium and 4 if it is large. We take the EU's EC recommendation as our reference, classifying firms according to the number of employees: micro (<10 workers); small (<49 workers), medium (from 50 to 250 workers) and large (more than 250 workers). Source for number of employees: SABI |
| R&D staff (EIDI) | discrete | Takes the value of 1 if the firm reports having R&D specialists. Takes the value of 0 otherwise. Source: SABI |
| Legal personality (PJ) | discrete | Legal personality of the firm: 1 if it is a cooperative, 0 otherwise. Source: SABI |
| Age (AGE) | continuous | The number of years the company has been active, calculated as the difference between the date of creation and the current date: 2017. Source: SABI |
| Index of economic activity (ECO) | discrete | Rate of growth of economic activity from 2008 to 2017 in the municipality where the firm is located. This is a dummy variable taking the value of 1 if growth is positive and 0 otherwise. Source: Based on data from the Institute of Statistics of Castilla-La Mancha |
| Rate of population growth (POB) | discrete | Rate of population growth from 2008 to 2017 in the municipality where the firm is located. It is a dummy variable taking the value of 1 if growth is positive and 0 otherwise. Source: Based on data from the Institute of Statistics of Castilla-La Mancha |
| Research Centers (CIDI) | continuous | Number of research centers per province in Castilla La Mancha. Source: Spanish national program for agri-food and forestry research and investigation |
| Level of education (EDUC) | continuous | Education index of population in Castilla-La Mancha. Source: La CAIXA |
| Area (Z) | discrete | 1: municipality in a regeneration area; 2 municipality in an intermediate rural area; 3 municipality in a peri-urban area. Source: Program for Sustainable Development of Rural Areas |

**Table 2.** Descriptive statistics for the variables in the model.

| | Minimum | Maximum | Mean | SD | Obs |
|---|---|---|---|---|---|
| **Continuous Variables** | | | | | |
| Innovation Index (II) | 0.0555556 | 1.555556 | 0.2873304 | 0.2309537 | 771 |
| Age | 1 | 93 | 26.24675 | 17.34598 | 771 |
| Research cents (CIDI) | 1 | 9 | 5.155642 | 2.604522 | 771 |
| Level of education (EDUC) | 0.69 | 3.11 | 2.457964 | 0.3994069 | 771 |

**Table 2.** *Cont.*

| Discrete Variables | | | | | |
|---|---|---|---|---|---|
| | Frequency 0 | Frequency 1 | Frequency 2 | Frequency 3 | Frequency 4 |
| Performance (ROA) | | 178 | 179 | 348 | |
| Size | | 430 | 264 | 70 | 6 |
| R&D staff (EDI) | 21 | 750 | | | |
| Legal form (PJ) | 587 | 184 | | | |
| Index of economic activity (ECO) | 4 | 767 | | | |
| Rate of population growth (POB) | 587 | 184 | | | |
| Area (Z) | | 132 | 353 | 229 | |

As shown, there is a wide range of variation in the innovation index, as confirmed by the Kruskal–Wallis test (chi-squared with ties = 9.176, $p$ = 0.0102), because the firms belong to different groups of agricultural activity. However, within the same group, firms have a similar II, with $p$ associated with the Kruskal–Wallis test in excess of $p > 0.05$: for primary agricultural products (chi-squared with ties = 102.000, $p$ = 0.4814); food processing (chi-squared with ties = 293.000, $p$ = 0.4890), and beverage processing (chi-squared with ties = 202.000, $p$ = 0.4868).

Finally, we used the GRETL, STATA 15, and SPSS 24 software to obtain statistical and econometric results.

### 2.3. Functional form of the Model

For this study, we used a Box–Cox regression model because dependent variable II does not follow normal distribution, according to the Shapiro–Wilk test with W = 0.89452 and $p$ = 0. We also considered a model for each type of agri-food activity: agricultural products, food processing and beverage processing, because the Kruskal–Wallis test showed differences in innovation between them.

The model is expressed analytically as follows:

$$II^{\theta}_i = \beta_0 + \beta_1 ROA_i + \beta_2 SIZE_i + \beta_3 EIDI_i + \beta_4 PJ_i + \beta_5 AGE_i^{\lambda} + \beta_6 ECO_i + \beta_7 POB_i + \beta_8 CIDI_i^{\lambda} + \beta_9 EDUC_i^{\lambda} + \beta_{10} Z_i + u_i$$

where $u_i \sim N(0, \sigma^2)$,

The dependent variable, innovation index (II), is subject to theta $\theta$ transformation. The independent variables were classified into two groups for internal and external factors, in line with the theoretical framework defined in Sections 1.1 and 1.2. The internal factors reflect firm structure: performance (ROA), size (SIZE), research personnel (EIDI), legal form (PJ), age (AGE). The second group covers the characteristics of the firm's location: economic activity (ECO), population (POB), research centres (CIDI), education level (EDUC), zoning (Z). The continuous independent variables of AGE, CIDI, EDUC are subject to lambda $\lambda$ transformation.

### 3. Results

Firstly, to determine if it was really necessary to apply a transformation ($\theta$, $\lambda$) to the dependent variable ($\theta$) and or the continuous independent variables ($\lambda$), the four-Box–Cox model procedure was applied (Table 3).

We chose the models with a $p$-value above 0.05, associated with the LR test for the $\theta$ and $\lambda$ powers with values ($-1$, 0, 1), and below 0.05 for the specific theta and lambda values (Table 4).

**Table 3.** Likelihood-Ratio (LR) statistic.

| | LR Statistic Test h0 | Restricted Log-Likelihood | LR Statistic chi2 | *p*-Value Prob > chi2 |
|---|---|---|---|---|
| **Primary Agricultural Products** | | | | |
| model(lhsonly) left-hand-side Box–Cox model | theta = −1 | 81.449844 | 3.93 | 0.047 |
| | theta = 0 | 78.452246 | 9.93 | 0.002 |
| | theta = 1 | 37.113568 | 92.61 | 0.000 |
| model(rhsonly) right-hand-side Box–Cox model | lambda = −1 | 37.843773 | 1.64 | 0.200 |
| | lambda = 0 | 37.400503 | 2.53 | 0.112 |
| | lambda = 1 | 37.113568 | 3.10 | 0.078 |
| model(lambda) both sides Box–Cox model with same parameter | lambda = −1 | 82.889397 | 3.26 | 0.041 |
| | lambda = 0 | 78.995324 | 11.05 | 0.001 |
| | lambda = 1 | 37.113568 | 94.81 | 0.000 |
| model(theta) both sides Box–Cox model with different parameters | theta = lambda = −1 | 82.889397 | 5.18 | 0.023 |
| | theta = lambda = 0 | 78.995324 | 12.97 | 0.000 |
| | theta = lambda = 1 | 37.113568 | 96.73 | 0.000 |
| **Food Processing** | | | | |
| model(lhsonly) left-hand-side Box–Cox model; | theta = −1 | 107.73331 | 49.56 | 0.000 |
| | theta = 0 | 128.43636 | 8.15 | 0.004 |
| | theta = 1 | 26.309 | 212.41 | 0.000 |
| model(rhsonly) right-hand-side Box–Cox model | lambda = −1 | 25.757771 | 2.65 | 0.104 |
| | lambda = 0 | 26.107326 | 1.95 | 0.163 |
| | lambda = 1 | 26.309 | 1.54 | 0.214 |
| model(lambda) both sides Box–Cox model with same parameter | lambda = −1 | 109.29841 | 48.18 | 0.000 |
| | lambda = 0 | 129.02715 | 8.73 | 0.003 |
| | lambda = 1 | 26.309 | 214.16 | 0.000 |
| model(theta) both sides Box–Cox model with different parameters | theta = lambda = −1 | 109.29841 | 49.15 | 0.000 |
| | theta = lambda = 0 | 129.02715 | 9.69 | 0.002 |
| | theta = lambda = 1 | 26.309 | 215.13 | 0.000 |
| **Beverage Processing** | | | | |
| model(lhsonly) left-hand-side Box–Cox model; | theta = −1 | 43.667467 | 61.32 | 0.000 |
| | theta = 0 | 74.063942 | 0.53 | 0.468 |
| | theta = 1 | 23.687199 | 101.28 | 0.000 |
| model(rhsonly) right-hand-side Box–Cox model | lambda = −1 | 23.668667 | 0.16 | 0.688 |
| | lambda = 0 | 23.746052 | 0.01 | 0.938 |
| | lambda = 1 | 23.687199 | 0.12 | 0.725 |
| model(lambda) both sides Box–Cox model with same parameter | lambda = −1 | 45.225122 | 59.52 | 0.000 |
| | lambda = 0 | 74.638303 | 0.70 | 0.404 |
| | lambda = 1 | 23.687199 | 102.60 | 0.000 |
| model(theta) both sides Box–Cox model with different parameters | theta = lambda = −1 | 45.225122 | 60.05 | 0.000 |
| | theta = lambda = 0 | 74.638303 | 1.22 | 0.269 |
| | theta = lambda = 1 | 23.687199 | 103.13 | 0.000 |

**Table 4.** Powers for Box–Cox procedure.

| | Power | Std. Coef. | Error | z | *p* > z |
|---|---|---|---|---|---|
| **Primary Agricultural Products** | | | | | |
| model(lhsonly) left-hand-side Box–Cox model | theta | −0.5839263 | 0.1996476 | −2.92 | 0.003 |
| model(rhsonly) right-hand-side Box–Cox model | lambda | −6.708896 | 9.03138 | −0.74 | 0.458 |
| model(lambda) both sides Box–Cox model with same parameter | lambda | −0.6210483 | 0.2012707 | −3.09 | 0.002 |
| model(theta) both sides Box–Cox model with | lambda | −3.999474 | 3.773758 | −1.06 | 0.289 |
| different parameters | theta | −0.5858607 | 0.1988411 | −2.95 | 0.003 |

**Table 4.** *Cont.*

| | Power | Std. Coef. | Error | z | *p* > z |
|---|---|---|---|---|---|
| **Food Processing** | | | | | |
| model(lhsonly) left-hand-side Box–Cox model | theta | −0.2669326 | 0.0960978 | −2.78 | 0.005 |
| model(rhsonly) right-hand-side Box–Cox model | lambda | 9.821598 | 9.442533 | 1.04 | 0.298 |
| model(lambda) both sides Box–Cox model with same parameter | lambda | −0.2767478 | 0.0963459 | −2.87 | 0.004 |
| model(theta) both sides Box–Cox model with | lambda | −4.615544 | 4.378688 | −1.05 | 0.292 |
| different parameters | theta | −0.272811 | 0.0961281 | −2.84 | 0.005 |
| **Beverage Processing** | | | | | |
| model(lhsonly) left-hand-side Box–Cox model | theta | −0.0806673 | 0.1116199 | −0.72 | 0.470 |
| model(rhsonly) right-hand-side Box–Cox model | lambda | 0.2047836 | 2.546802 | 0.08 | 0.936 |
| model(lambda) both sides Box–Cox model with same parameter | lambda | −0.0931118 | 0.1121316 | −0.83 | 0.406 |
| model(theta) both sides Box–Cox model with | lambda | −1.401793 | 1.781908 | −0.79 | 0.431 |
| different parameters | theta | −0.0937083 | 0.1117786 | −0.84 | 0.402 |

The Ramsey specification test was applied and, from the values that were correctly specified, we obtained the Adjusted Coefficient of Determination and the Root-Mean-Square Error, and selected the models with the largest Coefficient of Determination and the lowest Error (Table 5).

**Table 5.** Selection Models.

| | Test H0: | Ramsey RESET Test Ho: Model Has No Omitted Variables | Adj R-Squared | Root MSE |
|---|---|---|---|---|
| **Primary Agricultural Products** | | | | |
| model(lhsonly) left-hand-side Box–Cox model | theta = −0.5839263 | $F(3, 84) = 0.78$ Prob > F = 0.5070 | 0.1577 | 0.15637 |
| model(rhsonly) right-hand-side Box–Cox model | lambda = −1 | $F(3, 84) = 4.50$ Prob > F = 0.0057 | not applicable | not applicable |
| | lambda = 0 | $F(3, 84) = 4.79$ Prob > F = 0.0040 | not applicable | not applicable |
| | lambda = 1 | $F(3, 84) = 4.32$ Prob > F = 0.0071 | not applicable | not applicable |
| model(lambda) both sides Box–Cox model with same parameter | lambda = −0.6210483 | $F(3, 84) = 0.43$ Prob > F = 0.7330 | 0.1761 | 0.87249 |
| **Food Processing** | | | | |
| model(lhsonly) left-hand-side Box–Cox model | theta = −0.2669326 | $F(3, 257) = 0.53$ Prob > F = 0.6639 | 0.1558 | 0.12395 |
| model(rhsonly) right-hand-side Box–Cox model | lambda = −1 | $F(3, 257) = 2.03$ Prob > F = 0.1099 | not applicable | not applicable |
| | lambda = 0 | $F(3, 257) = 2.09$ Prob > F = 0.1017 | not applicable | not applicable |
| | lambda = 1 | $F(3, 257) = 2.62$ Prob > F = 0.0515 | not applicable | not applicable |
| model(lambda) both sides Box–Cox model with same parameter | lambda = −0.2767478 | $F(3, 257) = 2.11$ Prob > F = 0.0991 | not applicable | not applicable |

**Table 5.** *Cont.*

| | Test H0: | Ramsey RESET Test Ho: Model Has No Omitted Variables | Adj R-Squared | Root MSE |
|---|---|---|---|---|
| **Beverage Processing** | | | | |
| model(lhsonly) left-hand-side Box–Cox model | theta = 0 | F(3, 159) = 0.46<br>Prob > F = 0.7104 | 0.2987 | 0.626 |
| model(rhsonly) right-hand-side Box–Cox model | lambda = −1 | F(3, 159) = 1.27<br>Prob > F = 0.2857 | 0.2895 | 0.21808 |
| | lambda = 0 | F(3, 159) = 1.23<br>Prob > F = 0.2990 | 0.2901 | 0.21798 |
| | lambda = 1 | F(3, 159) = 1.16<br>Prob > F = 0.3272 | 0.2896 | 0.21805 |
| model(lambda) both sides Box–Cox model with same parameter | lambda = 0 | F(3, 159) = 0.63<br>Prob > F = 0.5954 | 0.3033 | 0.62393 |

The models estimated were found to be valid. F-Snedecor, with a *p*-value below 0.05, shows the global capacity of all the model's explanatory variables. These are models without multicollinearity, with a mean VIF below 10. The Breusch–Pagan test, with *p*-values above 0.05, shows the lack of heteroscedasticity in the models, so random disturbance is the same for all the observations. The residual sum of squares is close to zero (Table 6). The lack of correlation of the exogenous variables with the series of estimated residuals indicates that the independent variables do not present endogeneity problems when included in the models (Table 7).

**Table 6.** Estimation Results.

| | Primary Agricultural Products | Food Processing | Beverage Processing |
|---|---|---|---|
| ROA | 0.0024179<br>(0.12) | 0.0149154 *<br>(1.46) | 0.0251645<br>(0.40) |
| SIZE | 0.0833527 ***<br>(3.40) | 0.0699325 ***<br>(6.01) | 0.3907323 ***<br>(4.64) |
| EIDI | 0.1637163 **<br>(1.88) | 0.0057047<br>(0.15) | 0.5721744 **<br>(1.79) |
| PJ | 0.0813413 *<br>(1.38) | 0.0440443 *<br>(1.65) | 0.3146792 **<br>(1.85) |
| AGE | 0.0002473<br>(0.16) | 0.001354 **<br>(2.20) | 0.0069631 **<br>(1.96) |
| ECO | −0.0207900<br>(−0.5807) | 0.0850771<br>(0.94) | −0.323209<br>(−0.45) |
| POB | −0.0037742<br>(−0.23) | −0.00896325<br>(−0.4818) | 0.0523881<br>(0.38) |
| CIDI | 0.0025304<br>(0.36) | 0.00478769 *<br>(1.99) | −0.0201095<br>(−0.89) |
| EDUC | 0.0305965<br>(0.75) | 0.0256364<br>(1.10) | 0.3897629 ***<br>(3.22) |
| Z | 0.0117457<br>(0.60) | 0.0116547 *<br>(1.37) | 0.1229039 *<br>(1.54) |
| Cons | 0.152915<br>(0.93) | 0.3997562<br>(3.42) *** | 3.252276<br>(4.87) *** |
| F-Snedecor (*p*-value) | 3.26 (0.0018) | 7.13 (0) | 8.32 (0) |
| Residual sum of squares | 0.15637 | 0.12395 | 0.626 |
| Breusch–Pagan/Cook–Weisberg chi2(9) = 14.72; *p*-value | 14.72; 0.0990 | 22.85; 0.073 | 18.58; 0.0560 |
| Multicollinearity, Mean VIF | 1.18 | 1.17 | 1.60 |

t-statistics of the coefficient estimates in brackets, * Denotes significance at the 10-percent level; ** Denotes significance at the 5-percent level; *** Denotes significance at the 1-percent level.

**Table 7.** Endogeneity test: correlation between the residuals series and the exogenous variables.

| | Residuals (Primary Agricultural Products) | Residuals (Food Processing) | Residuals (Beverage Processing) |
|---|---|---|---|
| ROA | 0.000 | 0.000 | 0.000 |
| SIZE | 0.000 | 0.000 | 0.000 |
| EIDI | 0.000 | 0.000 | 0.000 |
| PJ | 0.000 | 0.000 | 0.000 |
| AGE | 0.000 | 0.000 | 0.000 |
| ECO | −0.0824 | 0.0505 | −0.0454 |
| POB | 0.000 | 0.000 | 0.000 |
| CIDI | 0.000 | 0.000 | 0.000 |
| EDUC | 0.000 | 0.000 | 0.000 |
| Z | 0.000 | 0.000 | 0.000 |

Finally, to measure the effect of each factor on innovation, like [102–104], we used the eta-squared test (Table 8) because it measures effect size. According to eta-squared, the greatest variability for innovation comes from firm size.

**Table 8.** Size Effect Statistics.

| Size Effect Eta-Squared | Primary Agricultural Products | Food Processing | Beverage Processing |
|---|---|---|---|
| ROA | 0.0001682 | 0.0081044 | 0.000974 |
| SIZE | 0.1173378 | 0.1218658 | 0.1171195 |
| EIDI | 0.0390329 | 0.000086 | 0.0193254 |
| PJ | 0.0214696 | 0.0103549 | 0.0207579 |
| AGE | 0.0002916 | 0.018195 | 0.023062 |
| ECO | 0.003587 | 0.0033906 | 0.0012737 |
| POB | 0.0001149 | 0.001846 | 0.0008706 |
| CIDI | 0.0014914 | 0.0150218 | 0.0048737 |
| EDUC | 0.0063991 | 0.0046068 | 0.0600394 |
| Z | 0.0040626 | 0.0071695 | 0.014371 |

## 4. Discussion

Hypothesis 1 (HP1), which poses that the degree of innovation in a firm is related to its characteristics, was confirmed, whatever the agri-food activity. This is in line with [7], which indicates that innovation stems from the firm's resources. Hypothesis 2 (HP2), which proposes that innovation behaviour depends on external factors, was partially confirmed because: (a) among the agri-food firms, only food and beverage processing firms were relevant; and (b) only research centres, level of education and location were significant [30,83].

If we focus on each type of activity, the estimations show the following. For firms with an innovative profile in the field of agricultural products, that is, large firms with more R&D+I personnel and a cooperative constitution, the external factors were not significant. For firms with an innovative profile in the field of food processing, that is, firms with greater profitability, larger size, more time in the sector and a cooperative constitution, it suits them to have research centres nearby and to be located in peri-urban areas, that is, areas with a growing population, medium-high income levels and situated close to urban or highly-populated areas. Finally, for innovative firms in the field of beverage processing, which tend to be larger, have more R&D+I personal, more time in the sector and a cooperative constitution, the most important external factors are areas with a higher education level and peri-urban location.

The common denominators of the three areas of activity in the agri-food sector are firm size and being a cooperative. These deductions are in line with studies that find a positive link between firm size and innovation [43,44,46,47,53,105]. However, they are not in line with others that find there may be no significant relationship between firm size

and propensity to innovate [58,106], or that large firms only invest in R&D when there are large-scale trading opportunities [49]. The cooperative legal form has been seen to be important for innovation [74,92]. This effect of the cooperative on innovation enables small producers to join up in order to gain the necessary size to innovate, thus becoming promoters of technical change thanks to the value of their long-term relations and the importance of the cooperative formation.

The common factors shared by food and beverage processing firms are age and firm location. Greater age has been found to have a positive effect on innovation [53,107]. However, other studies find a negative relationship because as firms age, they become less able to generate new innovations [58]. Location in less rural areas is also significant in processing firms, which confirms the importance of a less rural business fabric [31].

For food processing firms to innovate, it is important for them to be profitable: firms with a large business volume have more resources available so it is easier for them to take up potential business opportunities. This results in greater investment in R&D and more innovation [23,38,39]. For beverage processing firms, it is significant for them to have R&D+I personnel [92,106,108]. Some authors also state that knowledge should not only be obtained within the firm but should also be external, coming from outside research centres, universities, etc. [79–81,109].

Out of all these factors, the one that has the greatest influence on innovation in the agri-food sector, according to the eta-squared test, is firm size, for which the mean effect is 11%, far from that of the other determinations of innovation. As stated in [110], some authors find that firm size affects R&D+I activities because these need a large set of fixed and variable resources, but others consider that the flexibility and skill of small and medium enterprises allow them to compete and make up for the difference in scale when compared with large firms. It is clear that the other variables have very little effect on the degree of innovation in the agri-food industry in Castilla-La Mancha.

## 5. Conclusions

In the analysis performed on an inland region in southern Europe that specializes in the agri-food sector, it was shown that innovation activity by firms is mainly promoted by internal or firm factors. However, the factor that carries the greatest weight is firm size. The legal form should also be taken into account, as cooperatives become agents for change and promote innovation. Their democratic governance and participatory management help to develop trust-based networks that foster learning by doing, facilitate the circulation of information and develop incremental innovations. External factors also help, especially the level of education in the area, knowledge spillovers from research centres and location in less rural areas.

Even so, a certain weakness is found in the culture of innovation, which is reflected in the importance of the size variable. It is necessary to advance in territorial and social perception in more underprivileged areas so that innovation can become a priority and thus contribute to the development of the agrifood sector, because the innovation process is essential for growth in countries with an advanced economy and for balanced, sustainable development.

Support policies have proved to be necessary for facilitating innovation in the agrifood sector, especially for the decision on the initial investment. In our analyses, we found that regulation and public support for innovation were determining factors because a very large percentage of investment initiatives in agrifood firms received public support from the measures established (Focal and Innova), and access to public calls for application was widespread. A necessary future line of research is the efficiency of public regulation regarding ex-post knowledge spillover, and the potential for learning by doing for spatial differentiation of innovation in less privileged territories. Another future line of research that would consolidate the line we follow in our study would be to expand the sample of agrifood firms without focusing only on those with an innovation index above zero. This would give a better idea of innovative behaviour in general in the agrifood industry.

**Author Contributions:** Conceptualization: J.S.C.-V. Methodology, data collection and data analysis: M.C.G.-C. Data presentation, writing, reviewing and editing: J.S.C.-V. and M.C.G.-C. Both authors have read and agreed to the published version of the manuscript.

**Funding:** This research received no external funding.

**Institutional Review Board Statement:** Not applicable.

**Informed Consent Statement:** Not applicable.

**Data Availability Statement:** Data sets analysed during the study are available from the authors on reasonable request, although the data were obtained from the Iberian Balance Sheet Analysis System, Official Gazette of Castilla-La Mancha, Spanish Patent and Trademark Office, Institute of Statistics of Castilla-La Mancha and Program for Sustainable Development of Rural Areas, Caixa Bank.

**Conflicts of Interest:** The authors declare no conflict of interest.

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
