# Peer review of "Factors That Determine Innovation in Agrifood Firms"

_agronomy, doi:10.3390/agronomy11050989_

Round 1
Reviewer 1 Report
1) The title of the article does not reflect its content very well. The study analyzes the factors determining the innovativeness of companies in the agri-food industry. There is no question of any synergies.
2) The characteristics of the factors of innovation included in the research of other authors are very cursory. A lot of literature items are cited, but it is not known what results from the cited research.
3) Innovation in the agri-food industry is described very briefly. What innovations are you talking about? What purposes are they supposed to serve? Public aid is mentioned to support innovation in the studied region, but there is no general explanation for what the funds are directed to.
4) The assumption that regional aid for innovation, for which apply to the company is identified with their innovation is problematic. Therefore, the results can only be a contribution to a more in-depth analysis.
5) The algorithm of indicator IIi contains Spanish words.
6) The conclusions from the research are quite shallow. What conclusions can be directed to the policy of supporting innovation in the agri-food sector?
Author Response
Response to Reviewer 1 Comments
(The corrections that respond to comments by this Reviewer are shown in red in the main document)
Point 1: The title of the article does not reflect its content very well. The study analyzes the factors determining the innovativeness of companies in the agri-food industry. There is no question of any synergies

Response 1: We have changed the title in line with your suggestion: “Factors that determine innovation in firms in the agrifood sector”.
Point 2: The characteristics of the factors of innovation included in the research of other authors are very cursory. A lot of literature items are cited, but it is not known what results from the cited research
Response 2: We have completed section 1 (lines 28 to 35) and sections 1.1 and 1.2. (pp. 2, 3, 4).
Point 3: Innovation in the agri-food industry is described very briefly. What innovations are you talking about? What purposes are they supposed to serve? Public aid is mentioned to support innovation in the studied region, but there is no general explanation for what the funds are directed to.
Response 3: We are grateful for the comment and agree that it is necessary to clarify the components of the Innovation Index, which is made up of three elements:
- FOCAL refers to aid that is co-financed by the European Agricultural Fund for Rural Development (EAFRD) and contributions from the national and regional Administrations. It is offered to agrifood firms that wish to increase their added value by means of innovations affecting the processing and/or sale of their products.
- INNOVA, 80% co-financed by the EAFRD with the rest coming from national and regional Administrations. The aim is to promote R&D+I activities led by firms in order to boost their productivity and competitiveness by means of innovation projects to: a) develop, implement or set up new goods or services, or significantly improve those already produced or performed; and b) to develop and adopt new or significantly enhanced processes by adopting emerging technologies or applying new methods, standards or techniques in production, supply, logistics or organisation.
3) PATENTS AND UTILITY MODELS refer to technological innovations and their registration for legal protection, specifically in the Spanish Patent and Trade Mark Office (SPTO).
We have included these clarifications in Section 2.1. Measurement of innovation (p. 4)
Point 4: The assumption that regional aid for innovation, for which apply to the company is identified with their innovation is problematic. Therefore, the results can only be a contribution to a more in-depth analysis.
Response 4: The reviewer is correct in saying that reducing innovation to aid may be limited but in our analysis regulation and public support for innovation were found to be crucial. A very large percentage of investment initiatives in agrifood firms received public aid from the measures covered (Focal and Innova), and there was widespread access to public calls for applications.
Point 5: The algorithm of indicator IIi contains Spanish words.
Response 5: Thank you. We have corrected this.
Point 6: The conclusions from the research are quite shallow. What conclusions can be directed to the policy of supporting innovation in the agri-food sector?
Response 6: We have responded to this comment in the conclusions (pp 12-13) and are grateful for the opportunity to expand them.
Reviewer 2 Report
Review of the Manuscript
Synergies for Innovation in The Agri-Food Industry: Firm and Environment Factors
Submitted for publication to Agronomy MDPI
Summary of the paper
The Authors investigates the factors determining innovation in the agri-food sector in Castilla-La Mancha province of Spain. In the introduction, the Authors state the research question and list the internal and external factors affecting innovation. A concise literature review supports their choice. In the material and method section, they illustrate the empirical strategy, including the measurement of innovation, the dataset and the econometric approach. In the result section, the Authors described the empirical selection of the appropriate Cox-Box transformation based on the largest Coefficient of Determination and the lowest Root-Mean Square Error (table 5) and provided several specification tests to support their model selection. The Illustration of regression results and size effect statistics complete this part. In the discussion section, the Authors identify and comment the main drivers of innovation. They found that HP1 regarding the effect of internal factors was fully confirmed by data. HP2 about the effect of external factors was partially confirmed, because data supported a statistically significant effect only for 3 variables, only in the food and beverage processing sector. Firm size was the most important driver of innovation. In the conclusion section, the Authors provide a concise summary of the findings and limited policy advice.
Overall evaluation and key concerns
The paper is well-written and informative. It provides an interesting analysis of a key issue in the Agrifood system. The main contribution lies in the original dataset and in a comprehensive consideration of many internal and external factors. In this way, the Authors were able to explain what is really driving innovation in the area of study. The authors presented an in-depth empirical analysis to conclude on firm’s size being the topmost determinant of innovation in Agri-food sectors of the study region. Given that firm size contribution to innovation in Agrifood industry is an unsettled subject among academics, carrying out an empirical analysis helps to give clarity to the subject.
However, few important issues must be addressed before the manuscript can be considered for publication. I summarize the main concerns in the following four points:
- The Authors might consider explaining why the case study can be of interest for the broad, international audience of the journal. Although Authors provided exhaustive account of the importance of studying innovation drivers, they do not explain enough why the case of Castilla-La Mancha can provide international readers a valuable lesson. For example, in lines 35-38, it is reported the importance of the case study for Spanish agriculture and for the economy of the region. However, this is a convincing rationale for a regional journal, while it might be not enough for an international audience. The Authors might consider giving more emphasis to this point, in order to increase the impact and the citation potential of the article. Just a couple of sentences are enough explaining why the area is important for readers worldwide or what are the special characteristics that make this case unique and of interest.
- Material and methods. I have three points here that the Authors might want to clarify in the final version of the paper.
- Variation in the external drivers. To my best understanding, the external factors are measured using local data about Castilla-La Mancha. Then, as expected, all firms in the same area share the same values of the external drivers (e.g., all firms in the same area share the same rate of population growth, and the value of the variable POB is the same). Obviously, it is not possible that those data are computed for the same area and the same time interval, or there would not be variation in the data. I wonder if those variables refer to sub-areas of the Castilla-La Mancha region. In fact, if variation was obtained by changing the reference years across observation, there might be problems in measurement. The Authors might want to clarify this important point.
- Endogenous sample selection. The sample is limited to the firms who actually innovated in the period (?>0). This implies that the sample is not random and that a selection bias is possible (Heckman 1974 or any advanced econometric textbook). The Author might consider testing the opportunity for a two-step estimation procedure to correct this problem or provide a clear explanation of why this was not an issue. Note that this is the most important concern in the paper and – in my opinion – must be addressed before publication.
- Endogenous variables. The Authors might want to address the possible issue of endogeneity in the variable ROA. While it is true that profitability allows for R&D investments and therefore innovation, it is also true that innovation leads to higher profitability. Because the variable is computed as an average over the entire period, it is not possible to disentangle the two effects, and therefore endogeneity is possible. Authors might consider testing for endogeneity.
Minor remarks
Further considerations for revisions are:
- Required formatting of equations needed in Line 123 & 129.
- Repetition of N1 in Line 127. N1 should be taken out.
- Line 125 : Reasons for the time interval utilized should be stated.
- Table 1 should be formatted to make it easily readable.
- Take out “includes the” from Table 2 heading (Line 146).
- Take out the “W” after Shapiro-Wilk (Line 158).
- Line 166 - “[…] in line with the theoretical framework defined above”. Authors may consider stating the sections rather than using above.
- References must be formatted with journal titles abbreviated. Lines 274,276, 279, 280, 286, 288, 293, 295, 302 (just to mention a few), are improperly formatted according to the journal requirement for references.
Given these commends, I recommend the paper for consideration after major revisions.
Author Response
Response to Reviewer 2 Comments
(The corrections that respond to comments by this Reviewer are shown in blue in the main document)
Point 1: 
The Authors might consider explaining why the case study can be of interest for the broad, international audience of the journal. Although Authors provided exhaustive account of the importance of studying innovation drivers, they do not explain enough why the case of Castilla-La Mancha can provide international readers a valuable lesson. For example, in lines 35-38, it is reported the importance of the case study for Spanish agriculture and for the economy of the region. However, this is a convincing rationale for a regional journal, while it might be not enough for an international audience. The Authors might consider giving more emphasis to this point, in order to increase the impact and the citation potential of the article. Just a couple of sentences are enough explaining why the area is important for readers worldwide or what are the special characteristics that make this case unique and of interest.
Response 1: We are grateful to the reviewer for this suggestion and have stressed the importance of the agrifood sector in Castilla-La Mancha (lines 45 to 54, pp.1, 2)
Point 2: Variation in the external drivers. To my best understanding, the external factors are measured using local data about Castilla-La Mancha. Then, as expected, all firms in the same area share the same values of the external drivers (e.g., all firms in the same area share the same rate of population growth, and the value of the variable POB is the same). Obviously, it is not possible that those data are computed for the same area and the same time interval, or there would not be variation in the data. I wonder if those variables refer to sub-areas of the Castilla-La Mancha region. In fact, if variation was obtained by changing the reference years across observation, there might be problems in measurement. The Authors might want to clarify this important point.
Response 2: This has been clarified in Table 1: we take the municipal location of the firm. Therefore the indices for population and economic activity are those for their locations, during the period 2008-2017.
Point 3: Endogenous sample selection. The sample is limited to the firms who actually innovated in the period (?>0). This implies that the sample is not random and that a selection bias is possible (Heckman 1974 or any advanced econometric textbook). The Author might consider testing the opportunity for a two-step estimation procedure to correct this problem or provide a clear explanation of why this was not an issue. Note that this is the most important concern in the paper and – in my opinion – must be addressed before publication.
Response 3:
- It is correct that the firms that make up the data base are exclusively those that innovate because the goal was to focus on the factors that influence the degree of innovation in agrifood firms. In future research we could consider the option suggested by the reviewer with the aim of establishing differences between innovative and non-innovative firms. We are grateful for the suggestion.
- The firms that make up the data base are all the agrifood firms that requested public support, focusing on the FOCAL and INNOVA aid during the period 2008 to 2017. We are grateful for this comment and have made the necessary correction (lines 206 to 208, p.5).
Point 4: Endogenous variables. The Authors might want to address the possible issue of endogeneity in the variable ROA. While it is true that profitability allows for R&D investments and therefore innovation, it is also true that innovation leads to higher profitability. Because the variable is computed as an average over the entire period, it is not possible to disentangle the two effects, and therefore endogeneity is possible. Authors might consider testing for endogeneity.
Response 4: In statistics there is endogeneity when there is a correlation between the variable and the error term. We have therefore calculated the correlation between all the exogenous variables (including ROA) and the series of residuals resulting from each equation estimated. Both the ROA variable and the other exogenous variables are inter-correlated with the residuals, so there is no problem of endogeneity. In line with your suggestion, we have presented this in Table 7.
Point 5. Minor remarks
Further considerations for revisions are:
- Required formatting of equations needed in Line 123 & 129.
Response 5a: Corrected
- Repetition of N1 in Line 127. N1 should be taken out.
Response 5b: Corrected
- Line 125 : Reasons for the time interval utilized should be stated.
Response 5c: We chose this period of time because it covers the two rural development programmes for Castilla-La Mancha. Corrected.
- Table 1 should be formatted to make it easily readable.
Response 5d: corrected
- Take out “includes the” from Table 2 heading (Line 146).
Response 5e: corrected
- Take out the “W” after Shapiro-Wilk (Line 158).
Response 5f: Corrected
- Line 166 - “[…] in line with the theoretical framework defined above”. Authors may consider stating the sections rather than using above.
Response 5g: Corrected
- References must be formatted with journal titles abbreviated. Lines 274,276, 279, 280, 286, 288, 293, 295, 302 (just to mention a few), are improperly formatted according to the journal requirement for references.
Response 5h: Corrected
Round 2
Reviewer 2 Report
Thank you for addressing my comments. I understand that you want to leave the selection bias issues (Point 3 in my comments) for future research. Personally, I disagree with this choice. However, if the editor is willing to consider the paper anyway, you might want at least to acknowledge the limitation in a footnote or in a limitation/future research section of the conclusions.
I have no further comments
Author Response
Response to Reviewer 2 Comments
(The corrections that respond to comments by this Reviewer are marked in yellow in the main document)
Thank you for addressing my comments. I understand that you want to leave the selection bias issues (Point 3 in my comments) for future research. Personally, I disagree with this choice. However, if the editor is willing to consider the paper anyway, you might want at least to acknowledge the limitation in a footnote or in a limitation/future research section of the conclusions.
I have no further comments
Response: As already stated in the previous revision (point 3), this study is limited to firms that have an innovation index above zero, but we agree with the Reviewer that this amounts to a limitation for the research. Therefore, following the Reviewer’s suggestion, we have included in the Conclusions section a proposal for future research, indicating that our sample is limited because it focuses only on agrifood firms whose index is above zero (lines 354-357, p 13).
We are grateful to the Reviewer for the suggestion.